# Risk of aortic aneurysm and dissection following exposure to fluoroquinolones, common antibiotics, and febrile illness using a self-controlled case series study design: Retrospective analyses of three large healthcare databases in the US

**Ajit A. Londhe[1¤], Chantal E. Holy [2]\*, James Weaver [1], Sergio Fonseca[1], Angelina Villasis[1], Daniel Fife[1]**

1 Janssen Pharmaceutical Research and Development, LLC, Titusville, NJ, United States of America,
2 Johnson & Johnson, New Brunswick, NJ, United States of America

¤ Current address: Amgen, Thousand Oaks, CA, United States of America
* choly1@its.jnj.com

## Abstract

### Objective

Recent observational studies suggest increased aortic aneurysm or dissection (AAD) risk following fluoroquinolone (FQ) exposure but acknowledge potential for residual bias from unreported patient characteristics. The objective of our study is to evaluate the potential association between FQ, other common antibiotics and febrile illness with risk of AAD using a self-controlled case series (SCCS) study design.

### Design

Retrospective database analysis–SCCS.

### Setting

Primary and Secondary Care.

### Study population

51,898 patients across 3 US claims databases (IBM® MarketScan® commercial and Medicare databases, Optum Clinformatics).

### Exposure

FQ or other common antibiotics or febrile illness.

### Outcome

AAD.

**Data Availability Statement:** The data for these analyses were made available to the authors by third-party licenses from IBM MarketScan and Optum, commercial data providers in the US. The authors have a license for analysis of these data. Under the licensing agreement, the authors cannot provide raw data themselves. Other researchers could access the data by purchase through IBM/Optum, and the inclusion criteria specified in the Methods section would allow them to identify the same cohort of patients we used for these analyses. Future researchers may purchase the data using the following: - For the MarketScan databases: https://www.ibm.com/watson-health/products - For the Optum Database: https://www.optum.com/business/solutions/life-sciences/explore-data.html The authors of this research did not have special access privileges to the data that other researchers would not have.

**Funding:** This study was supported by Janssen Research & Development, LLC, who provided financial support for the work in the form of salaries for all the authors, and its commitment to pay the associated publishing fees. Janssen Research & Development, LLC's Standard Operating Procedures require publication of studies that, like this one, concern its products and were intended for publication when they were begun. For the present study, that intention is documented by its registration with ClinicalTrails.gov. The funding organization's Standard Operating Procedures also require that the study undergo a standard internal company review before it is submitted for publication. Internal company review is limited to roles not responsible for sales or marketing functions. Janssen Research & Development, LLC. did not have any additional role in the study design, data collection, analysis, or preparation of the manuscript. The specific roles of the authors are articulated in the 'author contributions' section.

**Competing interests:** At the time of the work, all authors were full-time employees of Janssen Research & Development, LLC, which is the Marketing Authorisation Holder for Levaquin (a fluoroquinolone) in the United States and some other countries. As full-time employees, the authors held stock or stock options. This does not alter the authors' adherence to PLOS ONE policies on sharing data and materials.

## Methods

We studied patients with exposures and AAD between 2012 and 2017 in 3 databases. Risk windows were defined as exposure period plus 30 days. Diagnostic analyses included p-value calibration to account for residual error using negative control exposures (NCE), and pre-exposure outcome analyses to evaluate exposure-outcome timing. The measure of association was the incidence rate ratio (IRR) comparing exposed and unexposed time.

## Results

Most NCEs produced effect estimates greater than the hypothetical null, indicating positive residual error; calibrated p (Cp) values were therefore used. The IRR following FQ exposure ranged from 1.13 (95% CI: 1.04–1.22 –Cp: 0.503) to 1.63 (95% CI: 1.45–1.84 –Cp: 0.329). An AAD event peak was identified 60 days before first FQ exposure, with IRR increasing between the 60- to 30- and 29- to 1-day pre-exposure periods. It is uncertain how much this pre-exposure AAD event peak reflects confounding versus increased antibiotic use after a surgical correction of AADs.

## Conclusion

This study does not confirm prior studies. Using Cp values to account for residual error, the observed FQ-AAD association cannot be interpreted as significant. Additionally, an AAD event surge in the 60 days before FQ exposure is consistent with confounding by indication, or increased use of antibiotics post-surgery.

## Registration

NCT03479736.

## Introduction

Aortic aneurysm (AA) is a condition affecting mostly elderly patients. Prevalence of abdominal AA–the most common form of AA–has recently been reported as approximately 2.2% of males and 0.4% of female population aged 65 and above [1, 2]. This condition is rarely present in patients less than 48 years of age [3]. The association between age and AA is due in part to the fact that AAs typically develop very slowly and are a result of multiple years of abnormal collagen content development [4].

Fluoroquinolones (FQ)–among the most commonly prescribed antibiotics in the United States–were found in animal studies to impair the quantity and quality of collagen production [5, 6]. The results of these preclinical studies have raised concerns that FQ may damage collagen in patients treated with FQ. Although most use of FQ in humans is of limited duration, exposure is common so the clinical risk of FQ for collagen-related adverse events has been extensively analyzed.

Four recent epidemiological studies have suggested an association between exposure to FQ and AA, aortic rupture, or aortic dissection (referred below in totality as "AAD") [7–10]. In a longitudinal cohort study, Daneman et al. compared risks of AA diagnoses during the 30 days after FQ or amoxicillin exposure vs the risks of AA during unexposed time and reported an adjusted hazard ratio (HR) for FQ of 2.24 (95% confidence interval [CI]: 2.02–2.49) and for

amoxicillin of 1.50 (95% CI: 1.32–1.70) [7]. Lee et al. used a nested case-control design in the Taiwan National Health Insurance Research Database from 2000 to 2011 to compare 1477 patients with AAD to 147,000 age and sex match controls and found an odds ratio of 2.43 (95% CI 1.83–3.22) for the association of AAD with current FQ use and 2.15 (95% CI 0.97–4.60) for the association of AAD requiring surgery and current FQ use. More recently, Lee et al. designed a case cross-over study that reported an adjusted odds ratio of 2.05 (95% CI 1.13–3.71) [9]. Pasternak et al. in a propensity-score matched cohort study in patients older than 50 in Sweden, found that the use of FQ increased the risk for AAD when compared to amoxicillin–hazard ratio 1.66 (95% CI 1.12–2.46)–resulting in an absolute number of 82 AAD cases per 1 million FQ prescription episodes [10]. A key limitation of these studies was their potential for bias. Confounding by indication, for example, when drugs were prescribed for a condition that may have led to AAD, was mentioned as a potential limitation. When patients were not used as their own controls, confounding due to unmeasured patient characteristics that vary little over time (such as smoking, obesity, exercise, or health behaviors) was also a potential limitation.

Another key limitation of prior analyses is the possibility of systemic bias. Residual bias can occur in all large retrospective database analyses after confounding control has been implemented and this bias can skew results in even the best designed studies. Approaches to identify residual bias often include the use of negative controls–exposures known to not cause the outcome of interest. The distribution of effects obtained from analyzing a large number of negative controls can be utilized to create a so-called calibrated p-value, one that, based on the data that we are actually using–rather than *a priori* statistical considerations–reflects the probability that the observed effect would be seen by chance [11–13]. For example, if the negative controls are not centered on the null value or are more scattered than expected, the calibrated p value would take this into account. Further details are added in the discussion section.

Our study was designed to evaluate the association between FQ and three distinct collagen-related adverse events: AAD, retinal detachment and Achilles tendon rupture. To address prior limitations of residual bias and systemic bias, we used a self-controlled case series study (SCCS) design and analyzed each outcome (AAD, retinal detachment and Achilles tendon rupture) for systemic bias using large sets of negative controls. This current paper focuses on the results of the AAD analyses. Results related to retinal detachment will be published in a subsequent publication. The analyses for Achilles tendon rupture could not be completed due to excess residual error, as observed with negative control analyses.

## Methods

All databases used in this study only contain de-identified patient data, no IRB approval was therefore required.

### Study design overview

We designed a SCCS study to compare FQ risk windows and non-risk windows within the same patients with AAD to estimate the relative incidence of the condition. This design protects against confounding by individual characteristics that may differ between patient groups in comparative study designs. Further, the method provides a mechanism to assess whether other biases are present (such as confounding by indication and protopathic biases) as it allows evaluation of risk both before and after exposures. We also estimated risk of AAD following exposure to commonly prescribed antibiotics (amoxicillin, azithromycin, trimethoprim with and without sulfamethoxazole) and febrile illness not treated with antibiotics (FINTA). These analyses were conducted to help contextualize the results from the FQ analysis. In addition, we

included diagnostic methodologies to assess residual error inherent to observational study designs by estimating risk of AAD from negative control exposures: exposures known not to be causally associated with AAD [13]. The distribution of effects from negative controls are used to look for evidence of bias and to empirically calibrate p values, which establishes a more realistic measure of statistical significance than is provided by traditional p values that reflect statistical variation but do not reflect bias. Finally, to increase generalizability of our findings, our study was replicated in 3 large US claims databases.

The study was pre-registered on clinicaltrials.gov as NCT03479736. It was designed to evaluate the association between FQ and three distinct collagen-related adverse events: AAD, retinal detachment and Achilles Tendon rupture. This current paper focuses on the results of the AAD analyses. Results for retinal detachment will be presented in a subsequent publication. Results for Achilles Tendon rupture could not be assessed due to excessive residual bias.

## Data sources

The following databases were used: IBM MarketScan® Commercial Database (IBMCOM), IBM Medicare® Supplemental Database (IBMMDCR), and Optum's de-identified ClinFormatics® Data Mart Database—Date of Death (OPTUMEXTDOD). The IBMCOM and IBMMDCR databases include patients with private insurance and together represent 147 million lives. The OPTUMEXTDOD database is also a US administrative health claims database and covers 82 million lives. The major data elements contained within these databases are outpatient pharmacy dispensing claims (coded with National Drug Codes [NDC]) as well as comprehensive listing of all inpatient and outpatient medical claims with procedure (coded in CPT-4, HCPCS, ICD-9-Proc or ICD-10-PCS) and diagnosis codes (coded in ICD-9-CM or ICD-10-CM). Because this study used anonymized data exclusively, it was exempt of IRB approval. Patients and the public were not involved in the design or conduct or reporting or dissemination of our study.

## Outcome definition

The following criteria were used to define an AAD event: an AAD event required presence of a primary diagnosis for aortic aneurysm, aortic rupture, or aortic dissection within a 2-week interval (+/- 1 week) relative to a surgical procedure for aortic repair in an inpatient or emergency department setting.

## Exposures and risk window

The study evaluated the effect of the following exposures on risk of AAD events: FQ medications (i.e., ciprofloxacin, gatifloxacin, levofloxacin, norfloxacin, moxifloxacin, gemifloxacin, or ofloxacin), amoxicillin, azithromycin, sulfamethoxazole with trimethoprim, sulfamethoxazole without trimethoprim, and FINTA. All drugs were identified by RxNorm codes for ingredient, thus all formulations were included in this study. For the drug exposures of interest: an exposure period was defined as the number of days of consecutive dispensing of drugs with no interruption of more than 30 days. FINTA was defined by a diagnosis of viral disease and of fever on the same day and no antibiotic prescriptions or inpatient admissions in the 60 days before or after the diagnosis. The exposure period was the number of consecutive days with distinct diagnoses of febrile illness. For all exposures of interest, the risk window for the primary analysis was defined as exposure period plus 30 days. This timing was selected based on findings from Daneman et al., who reported a mean 18 days (± 24.5 days) from fluoroquinolone initiation to tendon complications in post-marketing surveillance reports [7]. Non-risk

window (i.e., reference person-time) was defined as all other time periods after the naïve periods that were not in risk windows.

## Study population

Patients with at least 1 AAD event during the study period of April 1, 2012 to March 30, 2017 were initially included. From these patients, AAD events were retained for analysis provided they met the requirement of having 12 months of continuous medical and pharmacy benefit enrollment prior to the AAD event. The 12 months at the beginning of a patient enrollment period was considered the naïve period, a time during which exposures were ascertained but events excluded as not having the information required for analysis.

The study period end date of March 30, 2017 was selected because it was the most recent data available at the time of the analysis. The study period start date of April 1, 2012 ensured up to 5 years of data. As other FQ studies on the risk of collagen-related adverse events in similar databases were conducted until March 30, 2012 [14], this study provides a more recent population of AAD patients.

## Exclusion criteria

Patients were excluded from the study if they met any of the following exclusion criteria: 1) patients who experienced an AAD event while being within a risk window for multiple study exposures of interest; these patients were excluded because AAD attribution to a single study exposure could not be made. 2) Patients with inherited disorders of connective tissue, such as Ehlers-Danlos syndrome, epidermolysis bullosa, Marfan syndrome and osteogenesis imperfecta. 3) Patients with another collagen-related adverse event (Achilles tendon rupture or retinal detachment) during the 12-months prior to the AAD event.

## Sensitivity and post-hoc analyses

Sensitivity analyses included: 1) increasing the risk window to exposure duration plus 60 days, to allow comparison with other studies evaluating FQ and AAD [8, 10], and 2) modifying the definition of the AAD to include patients with diagnoses of AAD and diagnostic imaging for aneurysm in the 30 days prior to AAD diagnosis. In this sensitivity analysis, patients may or may not have received a repair procedure. Two post-hoc analyses were conducted on a subset of patients that did not have an inpatient hospitalization with a discharge date within 1) 30 days and 2) 60 days prior to AAD diagnosis. These post-hoc analyses were designed to exclude patients that may have had unobserved exposures because inpatient drug exposures are not captured in these US administrative claims databases.

## Sample size assessments

Sample size assessment was performed using the method described by Musonda et al. [15]. Using this methodology with standard type I and II error rates ($\alpha$ = 0.05, $\beta$ = 0.8), the sample sizes and event counts given the study design were powered to identify a minimum incidence rate ratio of 1.1.

## Statistical analyses

Incidence rate ratios (IRR) and 95% confidence intervals (CI) were calculated to quantify the relative incidence of AAD between risk windows and non-risk windows using a Poisson regression conditioned on the event [16]. To account for confounding by temporal factors that vary by age and season, linear combinations of cubic splines were modeled to approximate the

age and season effect and adjust the effect measure. Given the study population age ($>$ 50 years at event date) and a 5-year study window, 3 age knots were specified. 5 seasonality knots were specified to represent 4 seasons. Empirically calibrated p values were generated to account for residual random and systematic error, as described below. In contrast, the 95% CIs were nominally defined against the hypothetical null of IRR = 1. Results from each database were pooled using random-effects regression as described by Dersimonian and Laird [17] and only pooled estimates with heterogeneity ($I^2$) $<$ 50% were reported. When database-specific results showed $I^2 \geq$ 50%, estimates from each database were shown separately without pooling.

## Calibration of p values

To estimate residual error in each analysis, 38 exposures known to have no causal association with AAD were identified as negative controls [18]. These included: cyclobenzaprine, tramadol, benzonatate, pseudoephedrine, benzoyl peroxide, clobetasol, phenazopyridine, olopatadine, ascorbic acid, fluocinonide, antipyrine, dicyclomine, cefprozil, magnesium sulfate, terbinafine, terconazole, niacin, diphenoxylate, alendronate, permethrin, cetirizine, eszopiclone, oxybutynin, thiamine, phenobarbital, calcipotriene, sodium phosphate, acetic acid, pyrilamine, glucagon, exenatide, selenium sulfide, penciclovir, methylene blue, ciclesonide, clidinium, rifaximin, and loperamide. These negative control exposures were used to calibrate p values for the association between exposures of interest and AAD using the following methodology: the association between each negative control and AAD was estimated using the SCCS method and a distribution of those effects was generated. In absence of bias, negative controls in theory should produce precise effect estimates with 95% CIs encompassing 1.0. The observed distribution of the effect estimates of the 38 negative controls was defined as the empirical null distribution, which is interpreted as a measure of both the random and systematic error of the study design in conjunction with the observational data. The empirical null distribution was then used to calibrate p values to reflect the observed random and systematic error of the analysis, using methodologies described elsewhere [13].

## Pre-exposure analysis

The frequency of events was plotted during the 6 months before and 6 months after the first day of first exposure. This analysis provided an evaluation of the distribution of events relative to first exposure, both before and after. In addition, the IRR during two time periods before the first day of first exposure (risk window of: -60 to -30 days and -29 to -1 days, versus non-risk windows) were estimated. In addition, a pre-exposure analysis similar to the primary analysis was conducted, with the IRR of the 2 pre-exposure windows computed as covariates in the SCCS model.

## Concurrent drug analysis

The concurrent drugs analysis was similar to the primary analysis. However, for each exposure of interest, all other concurrent drug exposures were included in a regularized conditional Poisson regression to account for the potentially time-varying, confounding effects of multiple drug exposures and by proxy the conditions they treat [19].

## Results

Baseline Demographics: In OPTUMEXTDOD, IBMCOM and IBMMDCR, a total of 23,923, 16,667 and 11,308 cases were identified during the study period; amongst them, 43.13%,

**Table 1. Characteristics and exposure case count of distinct patients with AAD across each database.**

| DATABASE | OPTUMEXTDOD | IBMMDCR | IBMCOM |
|---|---|---|---|
| AAD cases (N) | 23,923 | 16,667 | 11,308 |
| Mean age (years) | 72 | 77 | 55 |
| Female (%) | 29 | 28 | 23 |
| Observation time (median) | 5.75 | 6.83 | 5.25 |
| Observation time (IQR) | 5.52 | 7.11 | 6.17 |
| Cases with FQ class exposure, N(%) | 10,319 (43.13%) | 8,757 (52.54%) | 3,181 (28.13%) |
| Cases with Febrile illness untreated with antibiotics exposure, N(%) | 423 (1.77%) | 124 (0.74%) | 183 (1.62%) |
| Cases with Amoxicillin exposure, N(%) | 8,008 (33.47%) | 6,682 (40.09%) | 5,252 (46.44%) |
| Cases with Azithromycin exposure, N(%) | 6,263 (26.18%) | 5,293 (31.76%) | 3,382 (29.91%) |
| Cases with Trimethoprim without Sulfamethoxazole exposure, N(%) | 324 (1.35%) | 327 (1.96%) | 133 (1.18%) |
| Cases with Trimethoprim with Sulfamethoxazole exposure, N(%) | 2,922 (12.21%) | 2,827 (16.96%) | 1,686 (14.91%) |

52.54% and 28.13% of all cases had exposures to FQ, respectively. Table 1 provides an overview of all characteristics of patients with AAD included in the study. For each exposure of interest, Table 1 shows counts of exposed AAD cases, however non-exposed AAD cases also contributed age- and season-specific event rate information to the model likelihood function.

## Empirical calibration

Across all databases, analyses using negative control exposures produced effect estimates for AAD that were on average greater than the hypothetical null (i.e., IRR = 1: in OPTUMEXTDOD: mean = 1.31, standard deviation (SD) = 1.24; in IBMCOM: mean = 1.21, SD = 1.35, in IBMMDCR: mean = 1.18, SD = 1.05) and suggestive of residual bias as shown in Fig 1. The empirical calibration diagnostic thus identified a moderate, positive residual bias that requires p values to be calibrated against the empirical null distribution that was greater than 1 for all analyses.

## Exposure timeline

The FQ exposure timeline for AAD across all 3 databases showed a similar pattern, where a surge in AAD was observed starting approximately 60 days before the first day of exposure and peaking between day -30 and day -1, as shown in Fig 2. The figure presents the frequency of AAD events (y-axis) over a 120-day period (60-days pre and 60-days post first exposure) centered on the day of first FQ exposure (x = 0). Red bars indicate that events occurred concurrently with the exposure (i.e., days during FQ exposure) whereas blue bars indicate events that occur outside of the risk window.

The IRR for AAD with FQ exposure in the pre-exposure intervals (-60 to -30 day and -29 to -1 day) were calculated in each database and are shown in Table 2. As expected from the exposure timeline charts, the IRRs are increased prior to exposure and were highest during the -29 to -1-day interval, in all databases, ranging from 3.45 (95% CI: 3.08–3.85) to 2.38 (95% CI: 2.22–2.55).

The exposure timelines for amoxicillin, azithromycin, trimethoprim with and without sulfamethoxazole and FINTA are shown in the S1–S3 Tables. Similar results with increased IRR prior to exposure were observed for amoxicillin and trimethoprim with sulfamethoxazole, in all 3 databases.

## Estimate of IRR for AAD in primary analysis

Estimates of IRR with 95% CIs and nominal and calibrated p values for AAD following exposures to FQ, other antibiotics, and FINTA are shown for each database in Table 3. For FQ, the

A: OPTUMEXTDOD

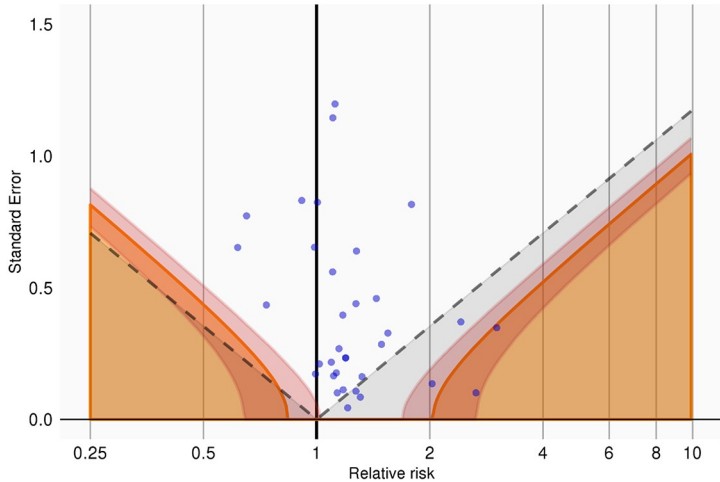

B: IBMMDCR

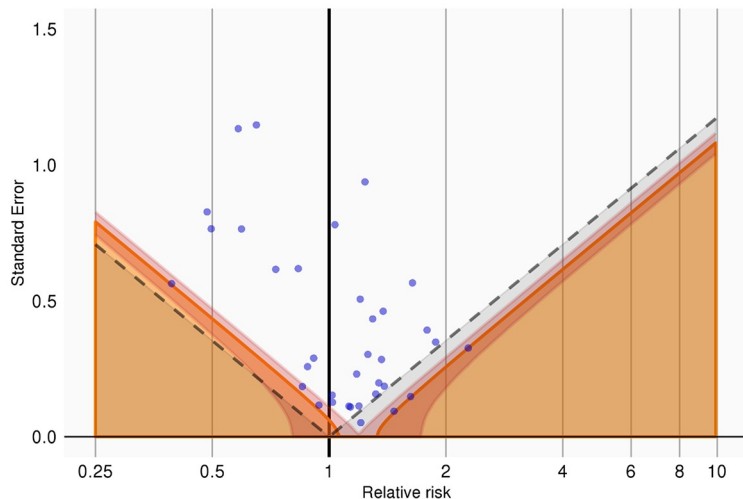

C: IBMCOM

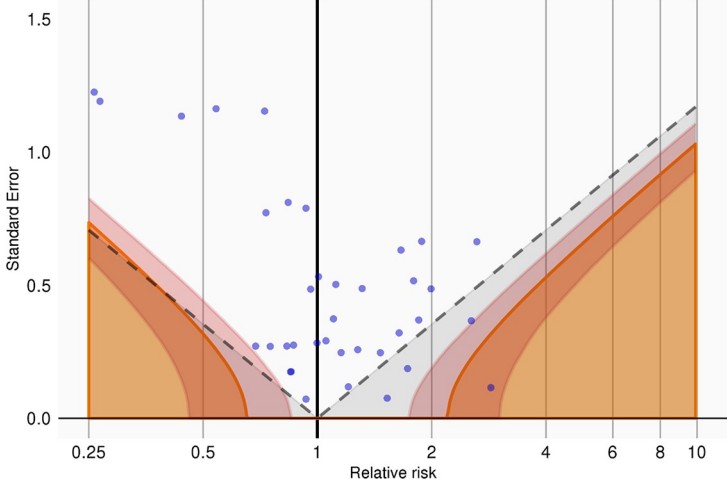

**Fig 1. Distribution of effect estimates for AAD for negative control exposures in each database.** A: OPTUMEXTDOD, B: IBMMDCR, and C: IBMCOM. Similar distributions were observed in all databases. Each blue dot represents the estimated incidence rate ratio and standard error (related to the width of the confidence interval) of each of the negative control exposures. Estimates below the dashed line have uncalibrated $p < 0.05$. In contrast, the estimates in the solid orange area have calibrated $p < 0.05$. The orange gradient indicates the 95% credible interval around the bold orange boundaries. That the effect estimates are not symmetrically distributed around the hypothetical null (IRR = 1) confirms the need for empirical calibration.

IRRs were elevated in all databases although none were statistically significant using calibrated p values. Pooled estimates were presented in all cases where $I^2$ values were < 0.5. Only the exposures to trimethoprim were sufficiently consistent across databases to permit pooling and their pooled estimates did not indicate increased IRR. None of the other exposures showed in any database an IRR that was both above 1 and statistically significant based on calibrated p-values.

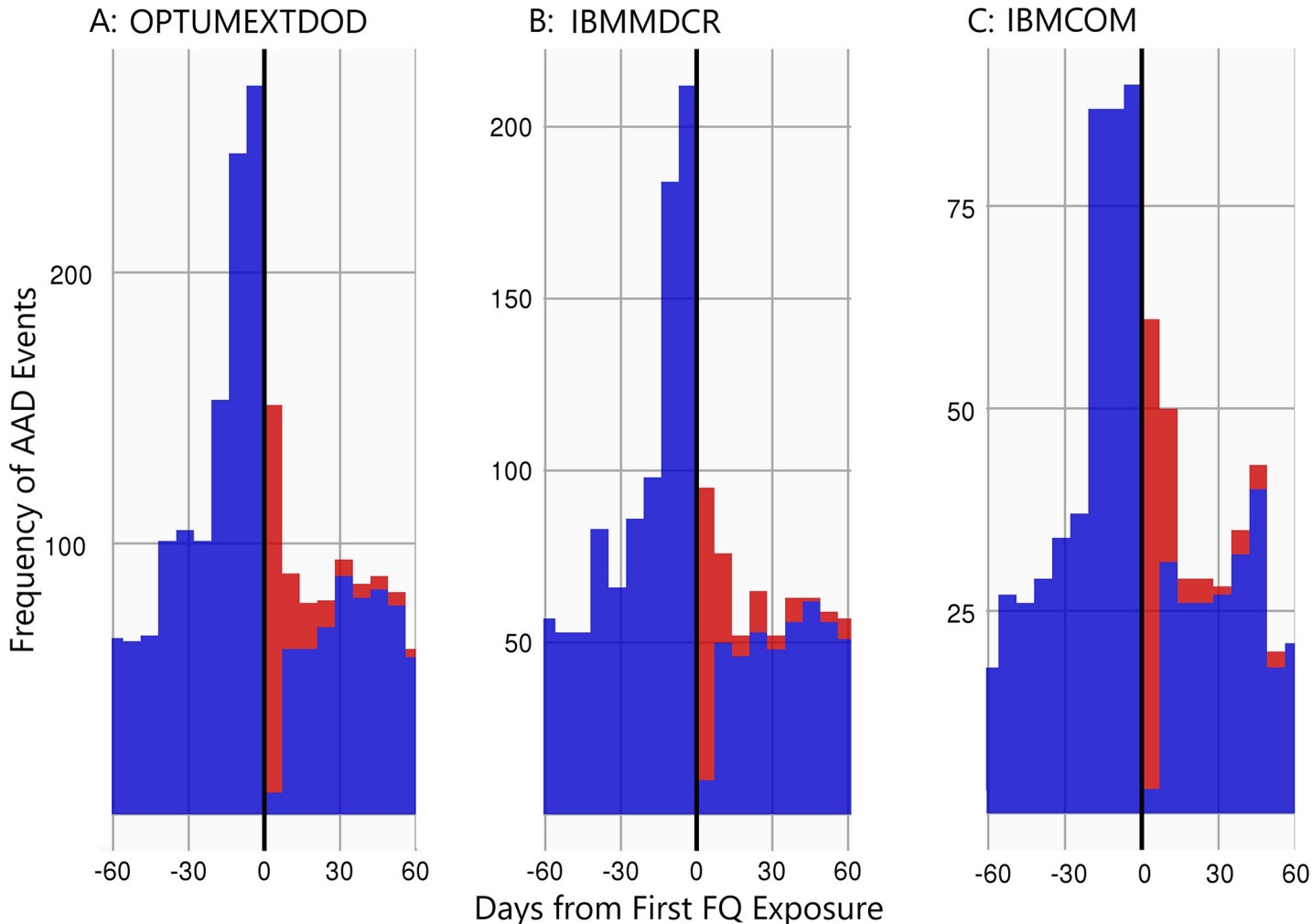

**Fig 2. FQ exposure timeline vs AAD events in each database.** A: Exposure timeline in OPTUMEXTDOD; B: Exposure timeline in IBMMDCR, C: Exposure timeline in IBMCOM. The exposure timeline figures present the frequency of AAD events (y-axis) over the 120-day period centered on the day of first exposure (x = 0). Red bars indicate that events occurred concurrently with the exposure window (i.e., days during antibiotics exposure, or FINTA, plus 30 days) whereas blue bars indicate events that occur outside of the risk window. A spike of events was observed prior to the first day of exposures to FQ.

**Table 2. Incidence rate ratios for AAD in two intervals prior to FQ exposure using SCCS design that is adjusted for seasonality and age, in each database.**

| FQ Exposure in the following Databases | IRR (-60d to -30d) | 95% CI LB (-60d to -30d) | 95% CI UB (-60d to -30d) | IRR (-29d to -1d) | 95% CI LB (-29d to -1d) | 95% CI UB (-29d to -1d) |
|---|---|---|---|---|---|---|
| OPTUMEXTDOD | **1.396** | 1.286 | 1.512 | **2.671** | 2.507 | 2.842 |
| IBMMDCR | **1.283** | 1.172 | 1.401 | **2.378** | 2.216 | 2.549 |
| IBMCOM | **1.483** | 1.266 | 1.726 | **3.452** | 3.085 | 3.852 |

Key: IRR = Incidence Rate Ratio, CI = Confidence Interval, LB = Lower Bound, UB = Upper Bound, d = days.

## Sensitivity analyses and post-hoc analyses

Sensitivity analyses were conducted to assess the robustness of each IRR to design specification changes for each exposure of interest. Sensitivity analysis IRRs with 95% CIs and nominal and calibrated p values are shown in Table 4. Although the IRR and CIs for AAD following exposure to FQ, FINTA, and other antibiotics showed some nominally increased and decreased risks, none of the effects were statistically significant following empirical calibration. Post-hoc analyses were also conducted using cohorts that excluded patients with inpatient care prior to events (to avoid exposure misclassification due to unknown exposures during inpatient

**Table 3. IRR with 95% confidence intervals and calibrated p values for AAD and following each exposure as indicated, for each of the databases, adjusted for age and seasonality.**

| | | IRR | 95% CI LB | 95%CI UB | p | C p |
|---|---|---|---|---|---|---|
| FQ | OPTUMEXTDOD | 1.242 | 1.159 | 1.329 | 0.000 | 0.797 |
| | IBMMDCR | 1.127 | 1.043 | 1.215 | 0.002 | 0.503 |
| | IBMCOM | 1.632 | 1.446 | 1.836 | 0.000 | 0.329 |
| | Pooled estimate: $I^2$ = 0.92 | | N/A | | | |
| Febrile illness not treated with antibiotics | OPTUMEXTDOD | 4.291 | 3.137 | 5.757 | 0.000 | 0.000 |
| | IBMMDCR | 1.532 | 0.592 | 3.245 | 0.326 | 0.561 |
| | IBMCOM | 0.709 | 0.217 | 1.692 | 0.511 | 0.391 |
| | Pooled estimate: $I^2$ = 0.86 | | N/A | | | |
| Amoxicillin | OPTUMEXTDOD | 1.002 | 0.918 | 1.091 | 0.969 | 0.246 |
| | IBMMDCR | 0.919 | 0.833 | 1.012 | 0.089 | 0.001 |
| | IBMCOM | 1.163 | 1.058 | 1.276 | 0.002 | 0.859 |
| | Pooled estimate: $I^2$ = 0.84 | | N/A | | | |
| Azithromycin | OPTUMEXTDOD | 1.153 | 1.039 | 1.276 | 0.007 | 0.578 |
| | IBMMDCR | 0.983 | 0.869 | 1.108 | 0.785 | 0.031 |
| | IBMCOM | 1.320 | 1.156 | 1.501 | 0.000 | 0.762 |
| | Pooled estimate: $I^2$ = 0.81 | | N/A | | | |
| Trimethoprim without Sulfamethoxazole | OPTUMEXTDOD | 0.706 | 0.398 | 1.159 | 0.201 | 0.082 |
| | IBMMDCR | 0.326 | 0.137 | 0.650 | 0.005 | 0.001 |
| | IBMCOM | 0.629 | 0.191 | 1.518 | 0.381 | 0.296 |
| | Pooled estimate: $I^2$ = 0.24 | | 0.55 (95%CI: 0.19–1.55) | | | |
| Trimethoprim with Sulfamethoxazole | OPTUMEXTDOD | 0.921 | 0.780 | 1.080 | 0.322 | 0.145 |
| | IBMMDCR | 1.067 | 0.906 | 1.247 | 0.428 | 0.317 |
| | IBMCOM | 1.065 | 0.874 | 1.287 | 0.521 | 0.720 |
| | Pooled estimate: $I^2$ = 0.00 | | 1.01 (95%CI: 0.82–1.25) | | | |

Key: IRR = Incidence Rate Ratio, CI = Confidence Interval, LB = Lower Bound, UB = Upper Bound, C p = Empirically Calibrated p value.

Calibrated p values can be larger or smaller than nominal p values depending on the width and central tendency of the empirical null distribution.

**Table 4. IRR for the sensitivity analyses for which risk period was defined as exposure with 60 days with 95% confidence intervals and calibrated p values for AAD and following each exposure as indicated, for each of the databases, adjusted for age and seasonality.**

|  |  | IRR | 95% CI LB | 95%CI UB | p | C p |
|---|---|---|---|---|---|---|
| FQ | OPTUMEXTDOD | 1.059 | 1.017 | 1.103 | 0.006 | 0.545 |
|  | IBMMDCR | 1.000 | 0.957 | 1.044 | 0.988 | 0.516 |
|  | IBMCOM | 1.316 | 1.216 | 1.422 | 0.000 | 0.867 |
| Febrile Illness not treated with antibiotics | OPTUMEXTDOD | 1.456 | 1.125 | 1.855 | 0.003 | 0.342 |
|  | IBMMDCR | 1.835 | 1.283 | 2.548 | 0.001 | 0.024 |
|  | IBMCOM | 1.341 | 0.858 | 1.997 | 0.173 | 0.873 |
| Amoxicillin | OPTUMEXTDOD | 0.900 | 0.855 | 0.946 | 0.000 | 0.133 |
|  | IBMMDCR | 0.861 | 0.816 | 0.908 | 0.000 | 0.097 |
|  | IBMCOM | 0.945 | 0.882 | 1.011 | 0.104 | 0.184 |
| Azithromycin | OPTUMEXTDOD | 0.918 | 0.865 | 0.973 | 0.004 | 0.164 |
|  | IBMMDCR | 0.813 | 0.761 | 0.868 | 0.000 | 0.042 |
|  | IBMCOM | 0.982 | 0.905 | 1.063 | 0.652 | 0.247 |
| Trimethoprim without Sulfamethoxazole | OPTUMEXTDOD | 0.753 | 0.558 | 0.996 | 0.055 | 0.053 |
|  | IBMMDCR | 0.626 | 0.451 | 0.845 | 0.003 | 0.009 |
|  | IBMCOM | 0.722 | 0.397 | 1.206 | 0.250 | 0.115 |
| Trimethoprim with Sulfamethoxazole | OPTUMEXTDOD | 0.873 | 0.792 | 0.961 | 0.006 | 0.104 |
|  | IBMMDCR | 0.967 | 0.879 | 1.060 | 0.475 | 0.401 |
|  | IBMCOM | 1.009 | 0.881 | 1.150 | 0.897 | 0.313 |

Key: IRR = Incidence Rate Ratio, CI = Confidence Interval, LB = Lower Bound, UB = Upper Bound, C p = Empirically Calibrated p value.

Calibrated p values can be larger or smaller than nominal p values depending on the width and central tendency of the empirical null distribution.

hospital stays). Results from the post-hoc analyses (S4–S12 Tables) and the SCCS including all other concurrent drugs (S13–S15 Tables) also supported the findings from the primary analyses shown above and are available in the Supplemental material. In these subsequent analyses, none of the IRRs for any of the exposures tested were significantly elevated after empirical calibration.

## Discussion

Our study was designed to evaluate the risk of AAD following exposure to FQ using the SCCS design that reduces confounding from unobserved patient characteristics. Additionally, we evaluated the risk of AAD following exposure to amoxicillin, azithromycin, trimethoprim with and without sulfamethoxazole, and FINTA to provide context to the FQ analysis by assessing risk in similar patients exposed to these non-FQ antibiotics. We empirically calibrated p values to reflect the distribution of IRRs obtained by executing the same study design against negative controls. This diagnostic step was critical in our study since the negative control effects showed considerable variability and were on average greater than 1 (i.e. the hypothetical null), suggesting a positively biased study design. We also repeated our analyses in 3 different US administrative claims databases–each comprising large populations–to increase the generalizability of our findings. Finally, we analyzed the timeline of event occurrences relative to the time of exposure. This analysis was central to understanding that the sequence of events in the patient's treatment pathway may artificially affect the risk of AAD. With a study design intended to address the limitations of prior research, the results suggest that none of the exposures evaluated herein are causally associated with an increased risk of AAD. For FQ, we found that the rate of AAD events increased significantly in the pre-exposure interval from 60- to -30 days, and more so in the interval from 29- to 1-day period before the first exposure to

FQ. This is consistent with a confounding factor that affects both AAD events and FQ exposure. However, such a peak is also consistent with increased use of antibiotics in the post-surgical period, after surgical correction of the AAD.

In contrast to this research, previous studies did not empirically assess how their reported limitations could affect the validity of their findings. Lee et al. [9] queried the Taiwan National Health Insurance Research Database from 2000 to 2011 to analyze the risks of AAD in patients taking oral FQ for at least 3 days using a case-control study design. The authors matched 1,477 cases hospitalized for AAD to 147,700 controls using propensity scores and reported a rate ratio (RR) of AAD for patients during FQ exposure of 2.43 (95% CI: 1.83–3.22), which decreased to 1.48 (95% CI: 1.18–1.86) for patients that discontinued FQ but were within a 60-day risk window of last FQ exposure. When restricting the analysis to patients that required surgery, the RR was estimated at 2.15 (95% CI: 0.97–4.60). Lee et al. noted as a key limitation the possibility of bias. A similar study also relying on propensity score matching was published by Pasternak et al. [10] that evaluated risk of AAD in Sweden in patients 50 years or older from 2006 to 2013. With no minimum number of days supply of prescription to define an exposure, Pasternak et al. evaluated risk within 60 days from start of treatment and compared the time to AAD events to patients treated with amoxicillin. The authors concluded that use of FQ was associated with an increased hazard of 1.66 (95% CI: 1.12–2.46) versus amoxicillin. Studies using propensity scores and other confounding adjustment strategies to compare separate patient groups cannot fully eliminate bias due to between-patient differences that are not well captured in databases, such as smoking, obesity, and other lifestyle characteristics that may differentially influence event occurrence. These lifestyle characteristics or behaviors are not well characterized in administrative claims databases and as such are left unaccounted in confounding adjustment.

A more recent analysis by Lee et al. [8] was published using the same data source as their 2015 manuscript, but used a case cross-over design, thus comparing within-patients whether FQ exposures were observed immediately preceding AAD events or during earlier time periods. This work included the same study population, outcome, and exposure definitions as the 2015 analysis but reported an odds ratio (OR) of FQ exposure in the AAD risk window of 2.71 (95% CI: 1.14–6.46) vs the prior non-risk window periods. A cohort study by Daneman et al., which used a patient population aged 65 years and older in Ontario, Canada from 1997 to 2012, utilized a different design and also attempted to reduce bias from unobserved patient characteristics. This study calculated the adjusted HR of AAD using FQ as a time-varying covariate and found an increased risk [HR = 2.24 (95% CI: 2.02–2.49)]. Using the same analysis, Daneman et al. found an increased but smaller risk of AAD following exposure to a negative control, amoxicillin [HR = 1.50 (95% CI: 1.32–1.70)]. The authors discussed the potential limitation that confounding by indication may be responsible for the reported effect. Both the Lee et al. (2018) and Daneman studies reduced risk of confounding from patient characteristics but were still not immune to confounding by indication or from residual error following their analyses.

There are several minor and major methodological differences between our study and these four previous analyses. For example, Lee et al. analyzed patients with a prescription of more than 3 days whereas our analysis included all patients with FQ exposure regardless of exposure period. However, the main differences concern our diagnostic analyses to address potential biases well known in observational research.

To evaluate the residual error of our observational study design, we employed negative controls methodology, which evaluated the risk of AAD from exposures known not to cause AAD or other collagen-related conditions [16]. In a perfect study design that fully accounts for selection effects, measurement error, and confounding, these exposures should be associated with

no risk of AAD (i.e. IRR = 1 with a low standard error). In our study, residual error was observed by analyzing the distribution of effects from the negative control exposures and we calibrated the p values accordingly [13]. It is important to note that p values should not be used for causal inferences–multiple publications have described limitations thereof [20]–but in our study, the use of calibrated p value was important to present the extent of calibration required to account for residual bias. In our study and as shown in Fig 1, the relative risks obtained from using negative controls were not centered around the null, as would be expected in a study with no residual bias. Negative control estimates were instead centered between 1 and 2, suggesting a moderate, positive systemic error. In Fig 1A–1C: the dashed lines represent thresholds of significance using non-calibrated p value: any estimate below the dashed lines, in a study with no residual bias, would be considered significant. In contrast, the estimates in the solid orange area have calibrated $p < 0.05$. Using non-calibrated p values, 8 out of 38 negative controls would have yielded significant risk estimates for AAD in OPTUMEXTDOD versus 2 out of 38 using calibrated p values. Calibrating p value was therefore essential to control for systemic error in our databases. None of the prior studies empirically calibrated their research designs against exposures or outcomes expected to produce a null result.

We also created exposure timelines, including AAD risk evaluation during the pre-exposure period, to understand the timing of events both before and after exposures. We observed a significant increase in AAD event rate in two time periods preceding FQ exposure. This peak of events is plausibly related to the fact that patients treated for AA are often prescribed antibiotics prophylactically after their AA repair surgery and before any dental, respiratory, gastrointestinal, genitourinary, dermatological, or musculoskeletal procedures [21]. However, such a peak may also suggest confounding. Our study consistently reported a peak in AAD events prior to exposure across 3 databases. None of the previous studies addressed AAD timing both before and after exposure.

Lastly, to evaluate generalizability of our findings, we reproduced our study in 3 large administrative claims databases: two that cover an employed, commercially insured population (IBMCOM and OPTUMEXTDOD) and one that covers retirees and supplementary beneficiaries (IBMMDCR). The consistency of results across multiple databases suggests heir generalizability.

There are several key limitations to our study. The SCCS study design requires that the probability of exposure be independent of occurrence of event. This requirement is difficult to ascertain in the context of antibiotic use and events that may possibly include prophylactic antibiotic prescriptions. In our study, a large pre-event peak in exposure could be due to prophylactic treatment of patients following AAD surgery, such that exposure–if that hypothesis is true–may not be independent of event. The presence of this peak could potentially also bias the IRR towards a lower risk. This observation could also explain why some antibiotics, in the Medicare database, showed a protective effect when no effect would have been anticipated. An additional potential bias may be related to prescriber awareness of potential detrimental effect of fluoroquinolones on AAD, thus affecting the independence between exposure and events. Furthermore, the use of negative controls for p value calibration, while described in multiple papers, is not currently considered standard and debate is ongoing regarding the most appropriate approaches to evaluate significance [11, 13, 22, 23]. In addition, the datasets used in this analyses (IBM and Optum) may have overlapping patients, complete independence between the databases cannot be ascertained. An additional limitation is the possibility of unknown confounders: though the SCCS study design allows for good control over fixed confounders, it does not eliminate the potential for all time-varying confounders. Finally, our study used prescription information from claims databases. These databases capture prescriptions as they are

filled by–and partially paid for–patients, but there is no certainty that patients actually took the drugs after filling the prescriptions.

In conclusion, our analyses did not confirm results from prior studies that suggest an association between FQ and AAD. Using empirical calibration to account for residual error, we found no statistically significant increased risk of AAD after exposure to fluoroquinolones.

## Supporting information

**S1 Table. Exposure timeline analysis: IRR estimates for AAD in OPTUMEXTDOD; risk window = exposures period + 30 days.**
(RTF)

**S2 Table. Exposure timeline analysis: IRR estimates for AAD in IBMMDCR; risk window = exposures period + 30 days.**
(RTF)

**S3 Table. Exposure timeline analysis: IRR estimates for AAD in IBMCOM; risk window = exposures period + 30 days.**
(RTF)

**S4 Table. Sensitivity analysis: IRR estimate for AAD in a subset of the primary population that did not have an inpatient hospitalization with a discharge date within 30 days of AAD.** Risk Window = Exposure period + 30 Days, Database = OPTUMEXTDOD.
(RTF)

**S5 Table. Sensitivity analysis: IRR estimate for AAD in a subset of the primary population that did not have an inpatient hospitalization with a discharge date within 30 days of AAD.** Risk Window = Exposure period + 30 Days, Database = IBMMDCR.
(RTF)

**S6 Table. Sensitivity analysis: IRR estimate for AAD in a subset of the primary population that did not have an inpatient hospitalization with a discharge date within 30 days of AAD.** Risk Window = Exposure period + 30 Days, Database = IBMCOM.
(RTF)

**S7 Table. Sensitivity analysis: IRR estimate for AAD in a subset of the primary population that did not have an inpatient hospitalization with a discharge date within 30 days of AAD.** Risk Window = Exposure period + 60 Days, Database = OPTUMEXTDOD.
(RTF)

**S8 Table. Sensitivity analysis: IRR estimate for AAD in a subset of the primary population that did not have an inpatient hospitalization with a discharge date within 30 days of AAD.** Risk Window = Exposure period + 60 Days, Database = IBMMDCR.
(RTF)

**S9 Table. Sensitivity analysis: IRR estimate for AAD in a subset of the primary population that did not have an inpatient hospitalization with a discharge date within 30 days of AAD.** Risk Window = Exposure period + 60 Days, Database = IBMCOM.
(RTF)

**S10 Table. Sensitivity analysis: IRR estimate for AAD in a subset of the primary population that did not have an inpatient hospitalization with a discharge date within 60 days of AAD.** Risk Window = Exposure period + 30 Days, Database = OPTUMEXTDOD.
(RTF)

**S11 Table. Sensitivity analysis: IRR estimate for AAD in a subset of the primary population that did not have an inpatient hospitalization with a discharge date within 60 days of AAD.** Risk Window = Exposure period + 30 Days, Database = IBMMDCR.
(RTF)

**S12 Table. Sensitivity analysis: IRR estimate for AAD in a subset of the primary population that did not have an inpatient hospitalization with a discharge date within 60 days of AAD.** Risk Window = Exposure period + 30 Days, Database = IBMCOM.
(RTF)

**S13 Table. Sensitivity analysis: IRR estimate for AAD, controlling for other concurrent drugs.** Risk Window = Exposure period + 30 Days, Database = OPTUMEXTDOD.
(RTF)

**S14 Table. Sensitivity analysis: IRR estimate for AAD, controlling for other concurrent drugs.** Risk Window = Exposure period + 30 Days, Database = IBMMDCR.
(RTF)

**S15 Table. Sensitivity analysis: IRR estimate for AAD, controlling for other concurrent drugs.** Risk Window = Exposure period + 30 Days, Database = IBMCOM.
(RTF)

## Acknowledgments

The authors wish to thank Dr. Li Lin for his editorial/writing support, and Ms. Prerna Kothari for programming support.

## Author Contributions

**Conceptualization:** Chantal E. Holy, Angelina Villasis.

**Formal analysis:** Ajit A. Londhe, Chantal E. Holy, James Weaver.

**Methodology:** Ajit A. Londhe, Chantal E. Holy, James Weaver, Sergio Fonseca, Angelina Villasis, Daniel Fife.

**Project administration:** Daniel Fife.

**Resources:** Sergio Fonseca.

**Software:** Ajit A. Londhe.

**Supervision:** Chantal E. Holy, Sergio Fonseca, Daniel Fife.

**Validation:** Ajit A. Londhe, Chantal E. Holy, James Weaver.

**Visualization:** Ajit A. Londhe, James Weaver.

**Writing – original draft:** Ajit A. Londhe, Chantal E. Holy, James Weaver, Sergio Fonseca, Angelina Villasis, Daniel Fife.

**Writing – review & editing:** Ajit A. Londhe, Chantal E. Holy, James Weaver, Sergio Fonseca, Angelina Villasis, Daniel Fife.

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
