## [Decision Letter · Decision Letter 0]

21 May 2021

PONE-D-21-04320

Risk of aortic aneurysm and dissection following exposure to fluoroquinolones, common antibiotics, and febrile illness using a self-controlled case series study design: retrospective analyses of three large healthcare databases in the US

PLOS ONE

Dear Dr. Chantal E Holy,

Thank you for submitting your manuscript to PLOS ONE. After careful consideration, we feel that it has merit but does not fully meet PLOS ONE’s publication criteria as it currently stands. Therefore, we invite you to submit a revised version of the manuscript that addresses the points raised during the review process.

We look forward to receiving your revised manuscript.

Kind regards,

Ping-Hsun Wu, M.D. PhD.

Academic Editor

PLOS ONE

Additional Editor Comments:

A more detailed explanation between calibrated p values and traditional p values is needed for the reader to understand their differences. Besides, it will be better to re-arrange the order of the findings in the result sections as the reviewer's suggestion.

Journal Requirements:

[All authors were full-time employees of Johnson & Johnson at the time of the study conduct. Licenses to databases was funded by Janssen Pharmaceuticals, a division of Johnson & Johnson. ].    

We note that one or more of the authors are employed by a commercial company: Johnson & Johnson and anssen Pharmaceuticals Research and Development LLC

Reviewers' comments:

Reviewer's Responses to Questions

**Comments to the Author**

1. Is the manuscript technically sound, and do the data support the conclusions?

Reviewer #1: Partly

2. Has the statistical analysis been performed appropriately and rigorously? 

Reviewer #1: Yes

3. Have the authors made all data underlying the findings in their manuscript fully available?

Reviewer #1: Yes

4. Is the manuscript presented in an intelligible fashion and written in standard English?

Reviewer #1: No

5. Review Comments to the Author

Reviewer #1: Summary

This study evaluated the potential association between fluoroquinolones (FQ), other common antibiotics (amoxicillin, azithromycin, trimethoprim), and febrile illness with the risk of aortic aneurysm or aortic dissection (AA/AD) using a self-controlled case-series design in three large US claims databases. The aim was to examine potential confounding in prior observational studies. Several issues listed below warrant further revision to improve study quality.

Comments

1. The authors calculated event occurrence before and after the first FQ exposure and estimated the incidence rate ratio (IRR) before the first FQ exposure to demonstrate there may be potential confounding for the association between FQ and AA/AD (Table 2 and Figure 2).

However, the timeline scales mentioned in the methods (e.g., within “60 days” before and after the first FQ exposure) and shown in the results (e.g., within “150 days” before and after the first FQ exposure in Figure 2) were not consistent, which may make the audience a little bit confusing. It will be better to have consistent timeline scales in the method and result sections.

2. The authors estimated calibrated p values for FQ, febrile illness not treated with antibiotics, amoxicillin, azithromycin, trimethoprim/sulfamethoxazole, and 38 negative exposure controls to demonstrate there may be bias or systematic errors for the association between FQ and AA/AD.

However, for the general audience, it is not easy or intuitive to understand why the measure of calibrated p values can indicate potential bias. It will be better to have more detailed explanation for the difference between calibrated p values and traditional p values and why calibrated p values can suggest potential bias in the methods section.

3. The authors mentioned they estimated IRR for FQ, febrile illness not treated with antibiotics, amoxicillin, azithromycin, and trimethoprim/sulfamethoxazole, followed by calibration of p values for 38 negative exposure controls and so called pre-exposure and timeline analyses. However, the results did not show up as this order, which made the article difficult to read and follow-up. It will be better to re-arrange the order of the findings (including main text, figures, and tables) in the result sections.

4. It will be better to indicate “Appendix table 1, Appendix table 2, and etc.” in the main text and in the supplemental materials, which will be better for the audience to find corresponding results.

6. PLOS authors have the option to publish the peer review history of their article (what does this mean?). If published, this will include your full peer review and any attached files.

Reviewer #1: No

---

## [Author Response · Author response to Decision Letter 0]

9 Jul 2021

Response to Reviewers

Additional Editor Comments:

A more detailed explanation between calibrated p values and traditional p values is needed for the reader to understand their differences. Besides, it will be better to re-arrange the order of the findings in the result sections as the reviewer's suggestion.

A paragraph explaining the negative control approach, and references to the p-value calibration work, had been added in the introduction: “Another key limitation of prior analyses is the possibility of systemic bias. Residual bias can occur in all large retrospective database analyses after confounding control has been implemented and this bias can skew results in even the best designed studies. Approaches to identify residual bias often include the use of negative controls – exposures known to not cause the outcome of interest. The distribution of effects obtained from analyzing a large number of negative controls can be utilized to create a so-called calibrated p-value, one that, based on the data that we are actually using – rather than a priori statistical considerations – reflects the probability that the observed effect would be seen by chance. (11-13) For example, if the negative controls are not centered on the null value or are more scattered than expected, the calibrated p value would take this into account. Further details are added in the discussion section.”

 The authors calculated event occurrence before and after the first FQ exposure and estimated the incidence rate ratio (IRR) before the first FQ exposure to demonstrate there may be potential confounding for the association between FQ and AA/AD (Table 2 and Figure 2).

However, the timeline scales mentioned in the methods (e.g., within “60 days” before and after the first FQ exposure) and shown in the results (e.g., within “150 days” before and after the first FQ exposure in Figure 2) were not consistent, which may make the audience a little bit confusing. It will be better to have consistent timeline scales in the method and result sections.

Figure 2 was modified to include only the 60-day pre- and post-exposure time period, to be consistent with the at-risk definitions described in the methods. 

2. The authors estimated calibrated p values for FQ, febrile illness not treated with antibiotics, amoxicillin, azithromycin, trimethoprim/sulfamethoxazole, and 38 negative exposure controls to demonstrate there may be bias or systematic errors for the association between FQ and AA/AD.

However, for the general audience, it is not easy or intuitive to understand why the measure of calibrated p values can indicate potential bias. It will be better to have more detailed explanation for the difference between calibrated p values and traditional p values and why calibrated p values can suggest potential bias in the methods section.

A paragraph explaining the methodology was added in the introduction (See above). In addition, greater discussion of our approach, and the rationale thereof, was added in the Discussion section: “In our study and as shown in Figure 1, the relative risks obtained from using negative controls were not centered around the null, as would be expected in a study with no residual bias. Negative control estimates were instead centered between 1 and 2, suggesting a moderate, positive systemic error. In Figures 1A-1C: the dashed lines represent thresholds of significance using non-calibrated p value: any estimate below the dashed lines, in a study with no residual bias, would be considered significant. In contrast, the estimates in the solid orange area have calibrated p < 0.05. Using non-calibrated p values, 8 out of 38 negative controls would have yielded significant risk estimates for AAD in OPTUMEXTDOD versus 2 out of 38 using calibrated p values. Calibrating p value was therefore essential to control for systemic error in our databases. “

3. The authors mentioned they estimated IRR for FQ, febrile illness not treated with antibiotics, amoxicillin, azithromycin, and trimethoprim/sulfamethoxazole, followed by calibration of p values for 38 negative exposure controls and so called pre-exposure and timeline analyses. However, the results did not show up as this order, which made the article difficult to read and follow-up. It will be better to re-arrange the order of the findings (including main text, figures, and tables) in the result sections.

The findings were reorganized accordingly.

4. It will be better to indicate “Appendix table 1, Appendix table 2, and etc.” in the main text and in the supplemental materials, which will be better for the audience to find corresponding results.

Appendix Tables were relabeled accordingly.

---

## [Decision Letter · Decision Letter 1]

27 Jul 2021

Risk of aortic aneurysm and dissection following exposure to fluoroquinolones, common antibiotics, and febrile illness using a self-controlled case series study design: retrospective analyses of three large healthcare databases in the US

PONE-D-21-04320R1

Dear Dr. Chantal E Holy,

We’re pleased to inform you that your manuscript has been judged scientifically suitable for publication and will be formally accepted for publication once it meets all outstanding technical requirements.

Kind regards,

Ping-Hsun Wu, M.D. PhD.

Academic Editor

PLOS ONE

Additional Editor Comments (optional):

All comments had been revised accordingly. This paper is suitable for publication.

Reviewers' comments:

Reviewer's Responses to Questions

**Comments to the Author**

1. If the authors have adequately addressed your comments raised in a previous round of review and you feel that this manuscript is now acceptable for publication, you may indicate that here to bypass the “Comments to the Author” section, enter your conflict of interest statement in the “Confidential to Editor” section, and submit your "Accept" recommendation.

Reviewer #1: All comments have been addressed

2. Is the manuscript technically sound, and do the data support the conclusions?

Reviewer #1: Yes

3. Has the statistical analysis been performed appropriately and rigorously? 

Reviewer #1: Yes

4. Have the authors made all data underlying the findings in their manuscript fully available?

Reviewer #1: Yes

5. Is the manuscript presented in an intelligible fashion and written in standard English?

Reviewer #1: Yes

6. Review Comments to the Author

Reviewer #1: In general, the authors responded the comments from the reviewers. However, it will be better to provide corresponding page number and line number in the main text for each response (e.g., calibrated P value), which facilitates the reviewers to confirm the questions raised have been well addressed.

7. PLOS authors have the option to publish the peer review history of their article (what does this mean?). If published, this will include your full peer review and any attached files.

Reviewer #1: No

---

## [Editor Report · Acceptance letter]

6 Aug 2021

PONE-D-21-04320R1 

Risk of aortic aneurysm and dissection following exposure to fluoroquinolones, common antibiotics, and febrile illness using a self-controlled case series study design: retrospective analyses of three large healthcare databases in the US 

Dear Dr. Holy:

I'm pleased to inform you that your manuscript has been deemed suitable for publication in PLOS ONE. Congratulations! Your manuscript is now with our production department. 

Kind regards, 

on behalf of

Dr. Ping-Hsun Wu 

Academic Editor

PLOS ONE